

# Benefits of Net Zero policies for future ozone pollution in China

Zhenze Liu[1,2], Oliver Wild[2], Ruth M. Doherty[3], Fiona M. O'Connor[4,5], and Steven T. Turnock[4,6]

[1]Jiangsu Key Laboratory of Atmospheric Environment Monitoring and Pollution Control, Collaborative Innovation Centre of Atmospheric Environment and Equipment Technology, School of Environmental Science and Engineering, Nanjing University of Information Science and Technology, Nanjing, China
[2]Lancaster Environment Centre, Lancaster University, Lancaster, UK
[3]School of GeoSciences, The University of Edinburgh, Edinburgh, UK
[4]Met Office Hadley Centre, Exeter, UK
[5]Department of Mathematics and Statistics, Global Systems Institute, University of Exeter, Exeter, UK
[6]University of Leeds Met Office Strategic Research Group, School of Earth and Environment, University of Leeds, Leeds, UK
**Correspondence:** Zhenze Liu (zhenze.liu@nuist.edu.cn)

**Abstract.** Net Zero emission policies principally target climate change, but may have a profound influence on surface ozone pollution. To investigate this, we use a chemistry-climate model to simulate surface ozone changes in China under a Net Zero pathway, and examine the different drivers that govern these changes. We find large monthly mean surface ozone decreases of up to 16 ppb in summer and small ozone decreases of 1 ppb in winter. Local emissions are shown to have the largest influence on

future ozone changes, outweighing the effects of changes in emissions outside China, changes in global methane concentrations and a warmer climate. Impacts of local and external emissions show strong seasonality, with the largest contributions to surface ozone in summer, while changes in global methane concentrations have a more uniform effect throughout the year. We find that while a warmer climate has a minor impact on ozone change compared to the Net Zero scenario, it will alter the spatial patterns of ozone in China, leading to ozone increases in the south and ozone decreases in the north. We also apply a deep

learning model to correct biases in our ozone simulations, and to provide a more robust assessment of ozone changes. We find that emission controls may lead to a surface ozone decrease of 5 ppb in summer. This is smaller than that simulated with the chemistry-climate model, reflecting overestimated ozone formation under present-day conditions. Nevertheless, this assessment clearly shows that the strict emission policies needed to reach Net Zero will have a major benefit in reducing surface ozone pollution and the occurrence of high ozone episodes, particularly in high-emission regions in China.

## 1 Introduction

Rapid changes in air pollution have occurred in China over the last few decades because of dramatic transformations in economic development and air pollutant emissions. Following substantial increases in emissions in the 1990s and 2000s, nationwide pollutant emission controls since 2013 have led to remarkable reductions in fine particular matter ($PM_{2.5}$), with national population-weighted annual mean concentrations decreasing from 62 to 42 $\mu$g/m$^3$ during 2013-2017 (Zhang et al., 2019).

However, surface ozone ($O_3$) pollution is becoming increasingly prevalent in China despite these emission controls, as recent emission policies have primarily targeted fine particles (Wang et al., 2022). Reductions in the emissions of nitrogen oxides



(NO$_x$), a precursor of both O$_3$ and fine particles, may lead to increased O$_3$ concentrations due to non-linear O$_3$ chemistry (Liu et al., 2021) and to strengthened incoming solar radiation (Hollaway et al., 2019). In addition, anthropogenic emissions of other O$_3$ precursors that contribute to O$_3$ formation e.g. volatile organic compounds (VOCs) and methane (CH$_4$) are less

well regulated (Li et al., 2019). There are also significant natural sources of O$_3$ precursors from vegetation and soils that may increase due to a warmer climate (Doherty et al., 2013; Fiore et al., 2015). Since surface O$_3$ is detrimental for human health, plant growth and crop yields (WHO, 2021), robust and effective emission controls on O$_3$ precursors are needed.

     The Intergovernmental Panel on Climate Change (IPCC) calls for cutting global greenhouse emissions to as close to zero to reduce the risks of climate change (IPCC, 2022). Many countries have recently adopted such Net Zero policies to reduce

net greenhouse gas emissions to zero by 2050, and China has also implemented emission policies that aim to achieve a carbon peak before 2030 and carbon neutrality by 2060 (Tay, 2022). These low-carbon policies alongside reductions in anthropogenic air pollutant emissions will have co-benefits for both global climate and air quality (UNEP, 2022). However, since surface O$_3$ changes are not directly proportional to emission changes, it is challenging to quantify the benefits for O$_3$ accurately. Future O$_3$ is also influenced by climate change through changes in atmospheric stagnation, natural emission sources, chemical reaction

rates, and deposition rates (Hong et al., 2019; Zanis et al., 2022; Brown et al., 2022). Regional surface O$_3$ changes also depend on emissions pathways in other parts of the world, which influence the long-range transport of O$_3$ and its precursors across continents (Wild et al., 2012; Doherty, 2015). The combination of these factors shapes the changes in future O$_3$ but imposes large uncertainties in O$_3$ projections (Turnock et al., 2020), which poses a challenge to assess the underlying impacts of Net Zero policies on future air quality.

While the general relationships between O$_3$, its precursor emissions and climate change are known well (Zeng et al., 2008; Hedegaard et al., 2013; Doherty et al., 2013; Griffiths et al., 2021), the relative importance of these drivers remains very uncertain. Challenges remain in the capability of chemistry-climate models to simulate O$_3$ changes accurately because processes occurring at small scales cannot be resolved adequately. Young et al. (2018) show that there are systematic biases in the simulation of present-day O$_3$ concentrations in current chemistry-climate models, and this raises questions over their skill in

representing long-term O$_3$ changes (Parrish et al., 2021). Averaging output from a number of different models is a common way to obtain more robust results, but does not eliminate the O$_3$ biases that are shown to be systematic (Revell et al., 2018). In addition, the models tend to use different parametrizations to represent different processes (Wild et al., 2020), and may misrepresent the importance of local emission controls or the risks caused by climate change. It is hence valuable to correct model simulations to produce more robust O$_3$ projections.

A practical way to address this is to apply deep learning models. Deep learning approaches have developed quickly in the last decade due to advances in computational speed that allow intensive training, and they have been applied widely in scientific fields (LeCun et al., 2015). Deep learning models have been shown to be a universal approximator (Hornik et al., 1989) and can thus be applied to compensate for discrepancies between physical model simulations and observations. We have demonstrated a successful application of deep learning to correct the biases in surface O$_3$ simulations from a global chemistry-climate model

(Liu et al., 2022a), and found that changes in surface O$_3$ in high-emission regions across the world may be overestimated with



the process-based model. This bias correction approach allows us to provide a more robust and reliable assessment of future surface $O_3$ projections under the effect of different emission policies and facilitates an examination of their effectiveness.

The aim of this study is to produce reliable estimates of future $O_3$ changes associated with changing emissions and climate under a Net Zero pathway in China, and to determine how well strict emission controls can tackle the increasing frequency
of high $O_3$ episodes. We introduce the chemistry-climate model used in Sect. 2 along with different emission and climate scenarios, and we describe the deep learning model that we have implemented to correct surface $O_3$ biases. We then investigate surface $O_3$ changes in China from the present day to the future under a Net Zero emission pathway in Sect. 3. The influences of emission changes outside China, changes in global $CH_4$ concentrations, and climate change are examined in Sect. 4. We demonstrate the capability of the deep learning model in simulating the biases in surface $O_3$, and apply this bias correction
technique to estimate future $O_3$ changes and high $O_3$ episodes in Sect. 5. Conclusions are presented in Sect. 6.

## 2   Approach

### 2.1   Description and application of the chemistry-climate model

We use version 1 of the United Kingdom Earth System Model, UKESM1 (Sellar et al., 2019) to simulate surface $O_3$ mixing ratios in the present-day (2013–2017) and the future (2060-2070) under different scenarios. UKESM1 consists of a physical
climate model, the Hadley Centre Global Environment Model version 3 (HadGEM-GC3.1), configured with the Global Atmosphere 7.1 and Global Land 7.0 (GA7.1/GL7.0) components (Walters et al., 2019), to which other Earth system processes are coupled (Sellar et al., 2019). A state-of-the-art module for modelling atmospheric composition in the troposphere and the stratosphere, the United Kingdom Chemistry and Aerosol model (UKCA; Morgenstern et al., 2009; O'Connor et al., 2014) is included. A gas-phase chemistry scheme, StratTrop (Archibald et al., 2020b) and an aerosol scheme, GLOMAP-mode (Mulc-
ahy et al., 2020) are used in UKCA. An extended chemistry scheme based on StratTrop that incorporates more reactive VOC species including alkenes, alkanes, and aromatics is used in this study to permit a more realistic representation of the chemical environment in China (Liu et al., 2021). The model resolution is N96L85 in the atmosphere, with $1.875°$ in longitude by $1.25°$ in latitude, 85 terrain-following hybrid height layers, and a model top at 85 km.

We use the atmosphere-only configuration of UKESM1 with prescribed present-day and future sea surface temperatures
(SST) and sea ice (SICE) in the form of monthly mean time-evolving fields to investigate the transient impacts of changing emissions under different climates. These fields alongside global values for greenhouse gas and methane concentrations are generated from fully coupled UKESM1 runs for historical and future simulations conducted as part of the Coupled-Model Intercomparison Project 6 (Eyring et al., 2016).

### 2.2   Emissions and experiments

We use CMIP6 year-2014 emissions, the latest year available, to represent present-day anthropogenic (Hoesly et al., 2018) and biomass burning emissions (Van Marle et al., 2017) for the globe, but replace anthropogenic emissions in China with



an up-to-date regional emission inventory over 2013-2017, the Multi-resolution Emission Inventory for China (MEIC; Li et al., 2017). Biogenic VOC emissions are calculated interactively with the iBVOC emissions scheme in the Joint UK Land Environmental Simulator (JULES) land-surface scheme (Pacifico et al., 2011), which is coupled to UKCA. Other online natural

emissions such as sea salt, dust and lightning $NO_x$ are the same as in UKESM1 simulations for CMIP6 (Turnock et al., 2020). Anthropogenic emissions for five sectors (industry, power plants, transport, residences, and agriculture) are provided for the model, and independent diurnal and vertical emission profiles are applied for each sector (Bieser et al., 2011; Mailler et al., 2013).

For the future, emissions under the shared socio-economic pathways (SSPs) of CMIP6 are used that account for future

social, economic, and environmental developments (O'Neill et al., 2014; Van Vuuren et al., 2014). We use the SSP1-1.9 pathway to represent Net Zero emission as net emissions of greenhouse gases drop down to zero at about 2060 in this scenario. We note that this scenario has the potential to limit global warming to 1.5 degrees Celsius by the end of this century. Future scenarios for China are taken from the Dynamic Projection model for Emissions in China (DPEC; Tong et al., 2020), and we use the "Ambitious pollution neutral goal" scenario to represent a net zero pathway in China. For comparison, we use the

SSP3-7.0 pathway from CMIP6, along with the corresponding "Baseline" scenario from DPEC, to represent a low mitigation scenario and to evaluate future $O_3$ pollution with high emissions. In addition, to assess the impacts of $CH_4$ on surface $O_3$, $CH_4$ concentrations from SSP1-1.9 and SSP3-7.0 are used to represent low and high $CH_4$ respectively.

We perform several model experiments to investigate surface $O_3$ changes and to quantify the contribution of emission changes inside and outside China, global $CH_4$ concentrations, changes in climate, see Table 1. For each of the future scenarios

the model is spun up for six years and then run for five years for data analysis. Table 2 summarises the global mean total surface emissions calculated from CMIP6, MEIC, and DPEC and the global $CH_4$ abundance.

**Table 1.** Model configurations used for the present-day (2013-2017) and six future (2060-2070) simulations. "Hist." means that the emissions, $CH_4$ concentrations or SST/SICE evolve as for the historical simulations. "NZ" means that they evolve under a net zero pathway. "High" means that they evolve under a high emission scenario, SSP3-7.0.

| Experiment | Emis. in China | Emis. outside China | $CH_4$ | SST/SICE |
|---|---|---|---|---|
| Present day | Hist. | Hist. | Hist. | Hist. |
| Net Zero | NZ | NZ | NZ | NZ |
| Local emis. | High | NZ | NZ | NZ |
| External emis. | NZ | High | NZ | NZ |
| High $CH_4$ | NZ | NZ | High | NZ |
| Warmer climate | NZ | NZ | NZ | High |
| SSP3-7.0 | High | High | High | High |



**Table 2.** Overview of annual mean time-varying surface emissions of $NO_x$, VOCs, CO from anthropogenic (ANT), biomass burning (BB), and biogenic (BIO) sources for the present day (2013–2017) and the future (2060–2070) Net Zero and SSP3-7.0 pathway in China. Annual mean surface $CH_4$ mixing ratios (ppb) are also shown.

| Emission (Tg(species)/yr) | | Present day | Net Zero | SSP3-7.0 |
|---|---|---|---|---|
| $NO_x$ | ANT | 24.2 | 2.9 | 33.9 |
| | BB | 0.3 | 0.2 | 0.3 |
| | Total | 24.5 | 3.1 | 34.2 |
| VOCs | ANT | 28.5 | 10.7 | 29.2 |
| | BB | 2.0 | 1.1 | 1.6 |
| | BIO | 38.0 | 56.4 | 56.9 |
| | Total | 68.5 | 68.2 | 87.6 |
| CO | ANT | 154.3 | 43.1 | 143.6 |
| | BB | 10.1 | 5.6 | 8.6 |
| | Total | 164.4 | 48.7 | 152.1 |
| $CH_4$ (ppb) | | 1844.4 | 1266.6 | 2733.5 |

## 2.3   Development of the deep learning model

A deep learning model is developed here to correct the biases in surface $O_3$ simulated with UKESM1. Like many other chemistry-climate models, UKESM1 exhibits systematic biases in surface $O_3$ (Turnock et al., 2020; Liu et al., 2022b; Archibald et al., 2020a), but it is hard to determine the origin of these biases. While some of these biases may be attributed to simplified chemistry, improvement in the chemical scheme in UKESM1 has been shown to increase biases in some locations (Archer-Nicholls et al., 2021). However, this problem can be addressed through deep learning to simulate the differences between the chemistry-climate model simulations and real-world observations. The model is trained on present-day conditions to establish a relationship between $O_3$ biases and key outputs of the chemistry-climate model, referred to as features. Future $O_3$ biases can then be predicted using features that are generated from simulations of the future with the chemistry-climate model. We adopt the approach applied by Liu et al. (2022a) to use meteorological and chemical variables as features. This approach has shown good performance in reproducing monthly mean surface $O_3$ biases over the globe, with a mean bias error of 0.1 ppb. In this study, we further develop and extend this deep learning model to predict the biases in daily mean $O_3$, which enables the examination of the occurrence of high $O_3$ episodes. We note that the $CH_4$ concentration is not included as an input feature because its variation under present-day conditions is much smaller than the changes expected in future. We therefore



adopt the non-linear parameterisation developed by Wild et al. (2012) to quantify the response of surface $O_3$ to changing $CH_4$ concentrations in future, and consider this feature independently of the others.

The Chinese air quality reanalysis dataset (CAQRA; Kong et al., 2021) assimilates hourly mean surface $O_3$ observations during 2013-2017 from the China National Environmental Monitoring Centre (CNEMC), and we use this as a reference to
derive surface $O_3$ biases in UKESM1 simulations. The surface $O_3$ reanalyses are shown to match observations well, with small mean errors of -2.3 $\mu g/m^3$ (Kong et al., 2021). We account for these errors and uncertainties, and represent them as noise which we add to the original dataset in model training. We assume that this noise follows a normal distribution with a mean of 2.3 $\mu g/m^3$ and one standard deviation of 2.3 $\mu g/m^3$, and generate three datasets with random noise to reduce overfitting in training. The CAQRA data at 15 × 15 km resolution are regridded to the coarser resolution of UKESM1. A key advantage
of the CAQRA data is that it provides complete spatial and temporal coverage for comparison with UKESM1, thus avoiding issues with the poor coverage of observations in some areas. However, we only examine data in areas below 2000 m altitude that have relatively high populations and where there are more measurement sites. For training, we pre-process the data to distribute them randomly across time and location, and then split them into a training set (80%), a validation set (10 %) and a testing set (10 %). The validation data are used to evaluate the model performance at each iteration of the training process, and
the test data provide an independent evaluation when the model training is completed.

## 3 Future surface $O_3$ changes in China under Net Zero policies

Seasonal mean surface $O_3$ mixing ratios in China simulated with UKESM1 are shown in Fig. 1 for the present day and the Net Zero pathway, without bias correction. There is a clear seasonal variation in surface $O_3$, with high summertime $O_3$ and low wintertime $O_3$ (Fig. 1a, d). However, this variation is reduced under Net Zero (Fig. 1b, e) due to $O_3$ decreases in summer (Fig.
1c) and $O_3$ increases in parts of eastern China in winter (Fig. 1f) in future. Surface $O_3$ mixing ratios decrease by about 16 ppb in summer, demonstrating the great benefits of emission controls in mitigating summertime $O_3$ pollution. However, smaller changes are seen in the most polluted industrial areas of China, namely the North China Plain, the Yangtze River Delta and the Pearl River Delta, even though reductions in anthropogenic emissions in these areas are substantially larger than other regions (Fig. S1a-b). This is principally due to VOC-limited $O_3$ formation regimes there in which decreased $NO_x$ emissions increase
$O_3$ mixing ratios (Liu et al., 2021). Much greater reductions in $NO_x$ emissions or further reductions in VOC emissions are needed to reduce surface $O_3$ mixing ratios in these high-emission regions. In contrast, higher emissions following SSP3-7.0 will greatly increase summertime $O_3$ (Fig. S2a-c), and the transport sector is shown to have the largest impact on $O_3$ changes.

In wintertime, there are decreases in surface $O_3$ mixing ratios in less polluted areas but increases in heavily populated industrial regions, and increases of up to 20 ppb occur in eastern China. This results in a reduced latitudinal gradient of $O_3$
mixing ratios in China in wintertime under the Net Zero scenario. These contrasting responses further demonstrate regional differences in the chemical environment for $O_3$ production. Polluted urban environments are dominated by VOC-limited $O_3$ formation, particularly in winter when weak boundary layer mixing permits greater $NO_x$ accumulation at the surface and rapid local $O_3$ destruction. Therefore, increased $NO_x$ emissions from the main emission sectors such as power plants, industry



and transport under SSP3-7.0 cause notable decreases in $O_3$ mixing ratios in winter (Fig. S2e-g) although the effect of the
residential sector is relatively small (Fig. S2h) as small changes in $NO_x$ emissions are accompanied by substantial changes in
VOC emissions (Cheng et al., 2021).

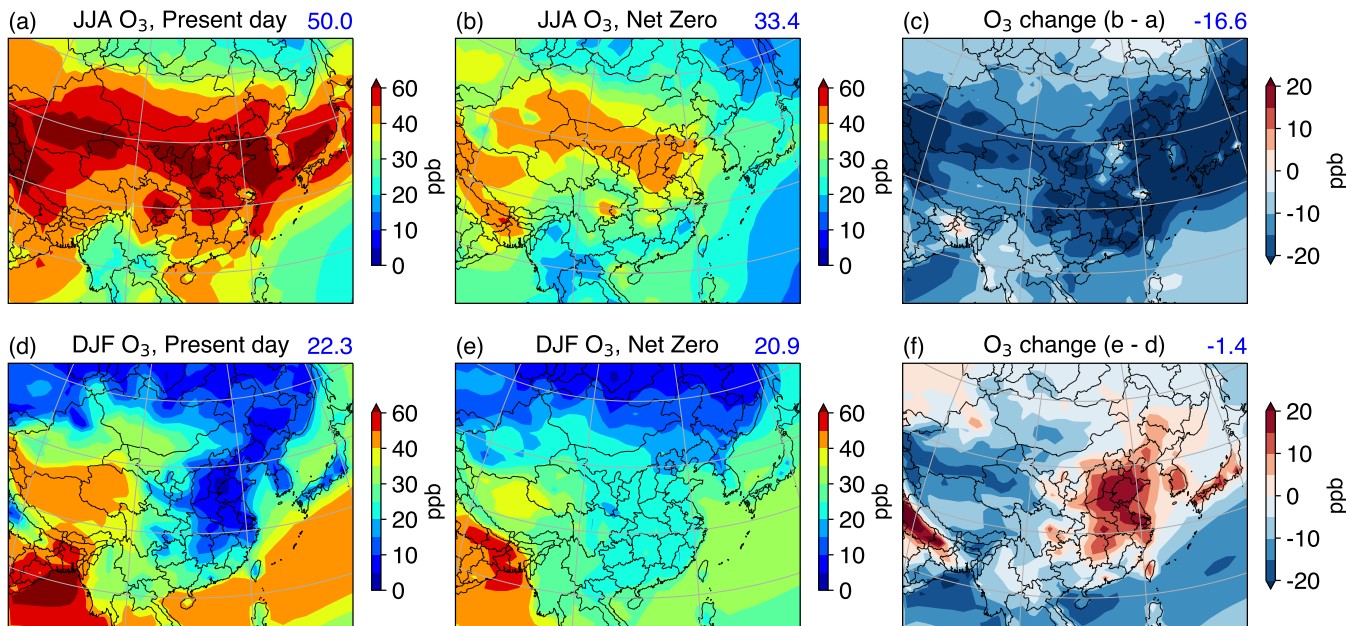

**Figure 1.** Seasonal surface $O_3$ mixing ratios in China simulated with UKESM1 from present day to the future following a Net Zero pathway.
Mean $O_3$ mixing ratios are shown for June-July-August (JJA; **a, b**) and December-January-February (DJF; **d, e**) along with the corresponding
seasonal changes **(c, f)**, with values of $O_3$ changes in China shown in ppb in the top right corner.

## 4   Drivers of future surface $O_3$ changes in China

While local emission changes directly influence surface $O_3$ changes in future, there are a number of other important drivers
that govern surface $O_3$. We investigate four independent drivers: changes in emissions inside (Local emis.) and outside China
(External emis.), changes in atmospheric $CH_4$ concentrations (High $CH_4$) and a warmer climate (Warmer climate) relative to
the Net Zero pathway, see Fig. 2. Local anthropogenic emission changes in China are shown to have the largest impact in
both seasons (Fig. 2a, e), but other drivers also contribute to surface $O_3$ changes and show substantial regional and seasonal
differences.

The effect of changes in emissions outside China reflects the importance of transport of $O_3$ from other countries and higher
background $O_3$ concentrations. If the rest of the world did not follow a Net Zero emission pathway, surface $O_3$ mixing ratios
would be more than 10 ppb higher in summer (Fig. 2b). The contribution to $O_3$ in winter is generally smaller, estimated here
as 4 ppb (Fig. 2f). Changes in atmospheric $CH_4$ abundance have a relatively uniform influence on surface $O_3$ in eastern China,




and lead to O$_3$ increases of 4 ppb in both seasons (Fig. 2c, g). The O$_3$ changes due to CH$_4$ are comparable to those across central China due to higher emissions outside China. In contrast, a warmer climate under the SSP3-7.0 scenario compared to the Net Zero pathway has minor impacts on surface O$_3$ changes (< 1 ppb). In general, surface O$_3$ mixing ratios decrease likely due to increased humidity under a warm climate but may increase locally due to higher temperatures, natural emissions and reduced O$_3$ deposition rates (Zanis et al., 2022). There are increased natural BVOC emissions in China under both Net Zero and SSP3-7.0 scenarios (Fig. S1c, f), particularly in southern China where vegetation is more abundant than in the north. Regional surface O$_3$ responds differently to different future climates (Fig. 2d, h), with O$_3$ increases in the south and O$_3$ decreases in the north under a warmer climate. These O$_3$ increases occur in both seasons but are more pronounced in summer. Overall, we show that while local emissions are critical to O$_3$ pollution, emissions outside China and global CH$_4$ concentrations are also important drivers of concern.

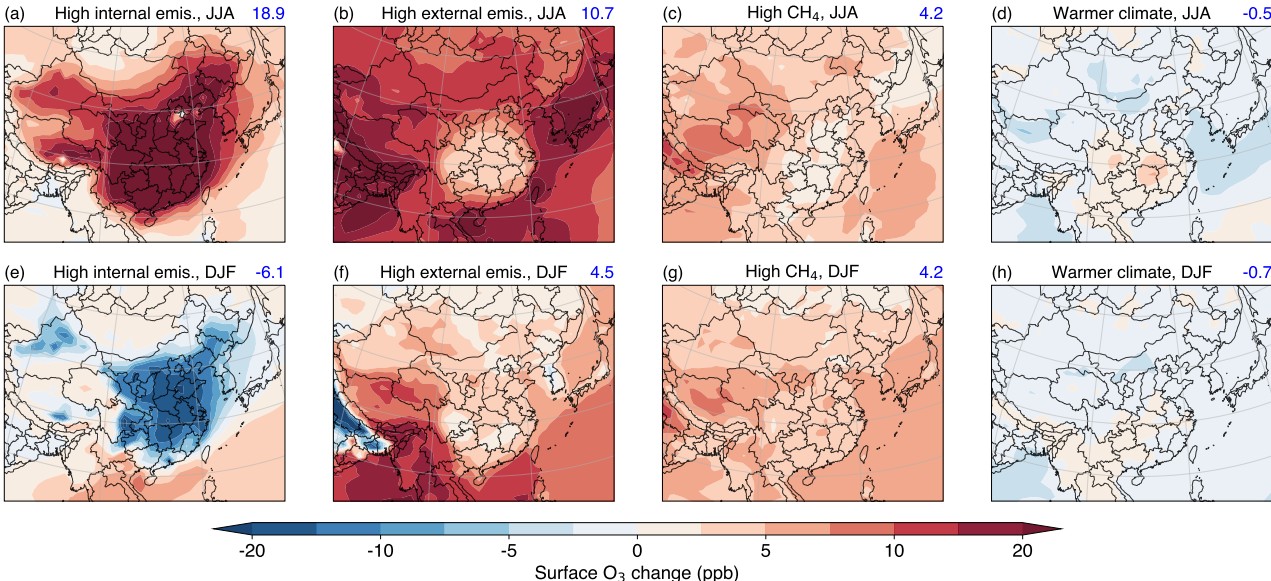

**Figure 2.** Contribution of changes in **(a)** internal emissions in China, **(b)** external emissions outside China, **(c)** global CH$_4$ concentrations and **(d)** a warmer climate following the SSP3-7.0 pathway to seasonal surface O$_3$ changes relative to the Net Zero pathway. Mean O$_3$ changes over China in ppb are shown in the top right corner.

The seasonality of surface O$_3$ changes in China and globally are shown in Fig 3. In summer, local emissions dominate surface O$_3$ increases, while in winter and spring, O$_3$ transport from other countries and O$_3$ increases due to elevated CH$_4$ concentrations are more important. Strong NO titration of O$_3$ leads to substantial O$_3$ decreases in winter, but its effects are suppressed by more efficient O$_3$ production over summer (Fig. 3a). Emissions outside China increase O$_3$ mixing ratios throughout the year, with the greatest impact in late spring and early summer when intercontinental transport is strongest. The seasonal variation in the influence of local and external emissions is relatively small on a global scale, reflecting a limited sensitivity of global O$_3$ changes to emissions (Fig. 3b). The uniform influence of changes in CH$_4$ concentration is comparable both in China and




globally. The warmer climate under SSP3-7.0 leads to slightly larger $O_3$ decreases on a global scale relative to the Net Zero scenario. We emphasize that seasonal $O_3$ responses to emission changes are more pronounced at a regional scale, and become weaker in winter, and that $O_3$ continental transport and background $O_3$ concentrations may still contribute to $O_3$ pollution.

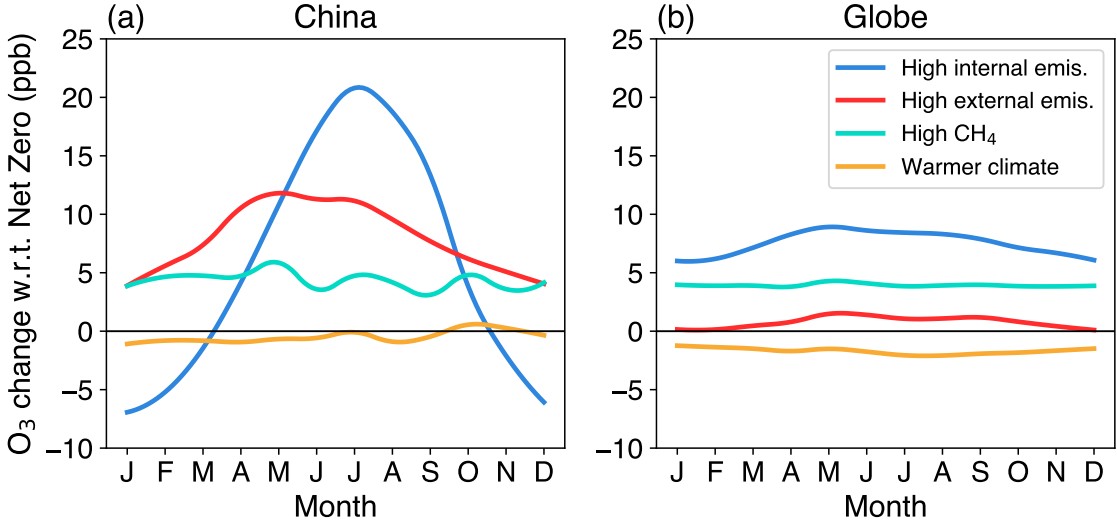

**Figure 3.** Seasonal surface $O_3$ changes relative to Net Zero due to changes in emissions in and outside China, global $CH_4$ concentrations and a warmer climate in **(a)** China and **(b)** the globe.

To examine how the occurrence of high $O_3$ episodes may change in future, we show the frequency distributions of daily mean surface $O_3$ mixing ratios for all grid cells over China under different scenarios in Fig. 4. We find that surface $O_3$ mixing

ratios under the Net Zero pathway follow an approximate normal distribution, with a mean $O_3$ of about 20 ppb (Fig. 4a). The frequency of high $O_3$ greater than 40 ppb can be greatly reduced under Net Zero. This is substantially different from the present day and SSP3-7.0 scenarios. SSP3-7.0 assumes that there are no emission controls in China, leading to a higher frequency of high $O_3$ mixing ratios (> 50 ppb). However, the faster NO titration on $O_3$ with higher $NO_x$ emissions also increases the frequency of low $O_3$ mixing ratios (< 10 ppb). In Fig. 4b, we show that the $O_3$ distribution shifts to higher values of $O_3$ under

the high internal emission scenario and is substantially different from the other scenarios shown here, indicating that there is a large change in local $O_3$ production due to local emission changes. The frequency of $O_3$ mixing ratios between 30 and 50 ppb are highest in the scenarios of high external emissions and high $CH_4$ concentrations, demonstrating that these factors can lead to an overall increase in daily mean $O_3$. In addition, we do not find significant changes in $O_3$ mixing ratios due to a warmer climate under SSP3-7.0.



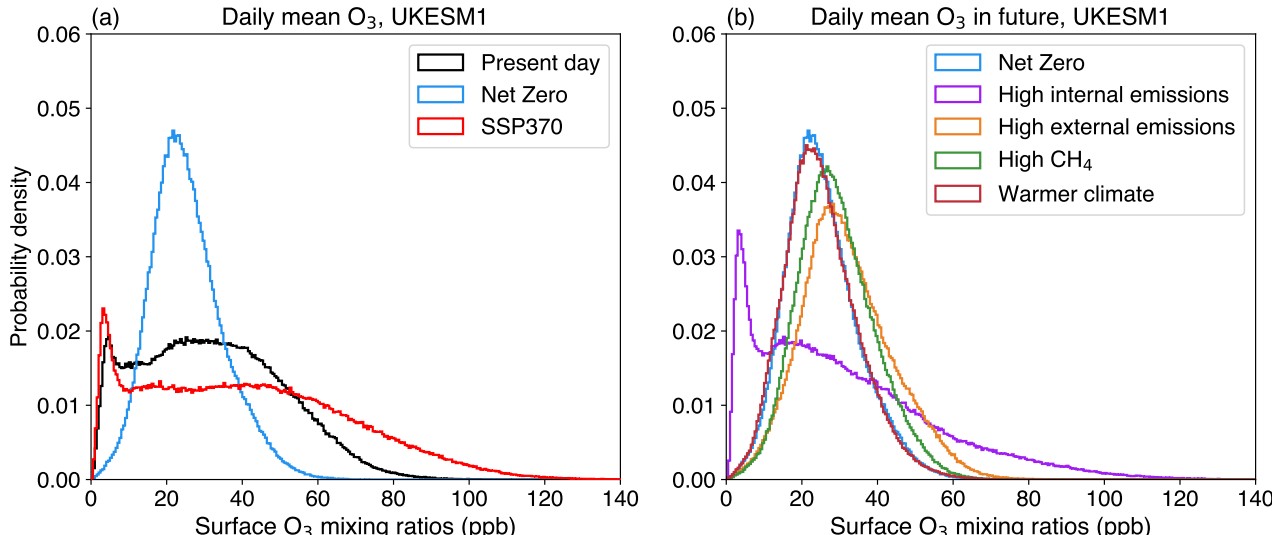

**Figure 4.** Whole year distributions of daily mean surface $O_3$ mixing ratios **(a)** in the present day, the Net Zero and the SSP3-7.0 scenarios in China, and **(b)** in the scenarios with higher internal emissions, external emissions, $CH_4$ concentrations and a warmer climate relative to Net Zero.

## 5  Bias corrected surface $O_3$ under the Net Zero pathway

Since there are systematic biases in surface $O_3$ simulations with UKESM1 (Fig. S3a, b), the reliability of future $O_3$ projections remains uncertain. We estimate the biases in surface $O_3$ through the deep learning model, and apply this to generate a more robust assessment of $O_3$ changes under the Net Zero pathway. A fully independent evaluation for the deep learning model is conducted using a testing dataset, see Fig. 5. We show that the magnitudes and distributions of biases in the UKESM1 simulations are reproduced well by the deep learning model, with a correlation coefficient of 0.96, a mean bias error of 0.1 ppb and a root-mean-square error of 4.0 ppb, which demonstrates the robustness of this approach. We also subtract the biases from UKESM1 and examine the spatial and temporal distribution of $O_3$ mixing ratios in China in Fig. 6. Spatial distributions of surface $O_3$ in China over 2013-2017 can be also captured well (Fig. 6a, b, d, e), with the highest summertime $O_3$ and the lowest wintertime $O_3$ in the North China Plain. The magnitudes of surface $O_3$ mixing ratios with bias correction are in close agreement to the observations. The time series of daily mean $O_3$ can be also simulated well in Beijing and Guangzhou (Fig. 6c, f), which represent two different locations in northern and southern China with rather different chemical and meteorological environments. The evaluation demonstrates the capability of the deep learning model in correcting the seasonal and daily UKESM1 simulation of surface $O_3$.

Spatial distributions of future bias-corrected surface $O_3$ under the Net Zero pathway are shown in Fig. 7 to compare and contrast with UKESM1 outputs (Fig. 1), and to assess the effectiveness of emission controls. With bias correction, summertime $O_3$ mixing ratios generally decrease under Net Zero (Fig. 7a, b), consistent with UKESM1 results (Fig. 1c). We find that there



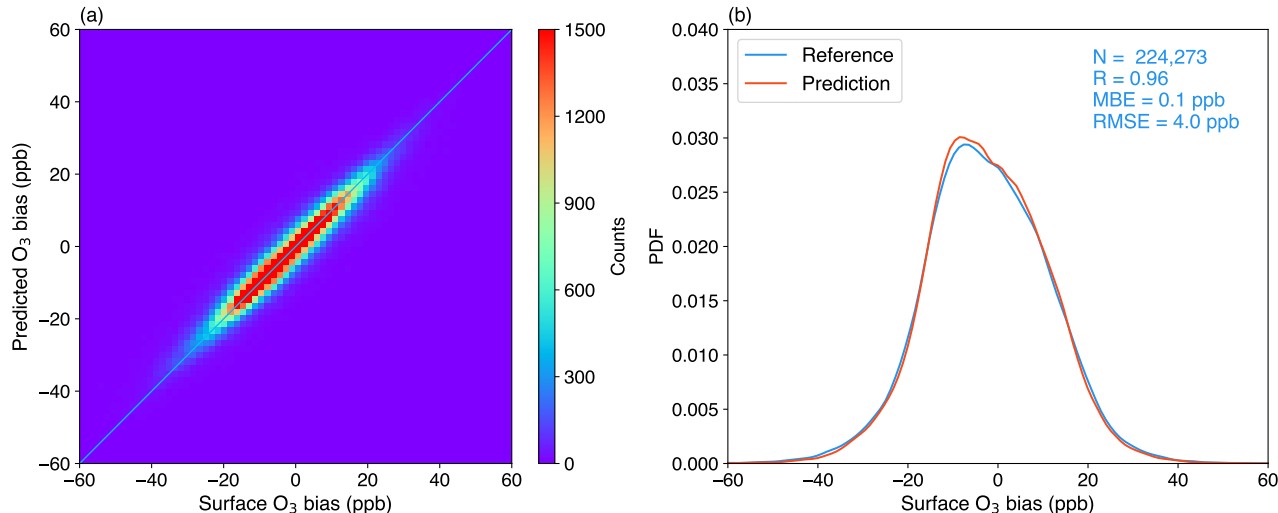

**Figure 5.** Independent evaluation of the deep learning model in simulating daily mean surface $O_3$ biases at each UKESM1 grid point over China. **(a)** Surface $O_3$ biases (UKESM1 minus CAQRA) and biases predicted by the deep learning model. **(b)** Probability density function (PDF) of $O_3$ biases (labelled here as Reference) and predicted $O_3$ biases. Statistics are shown in the top right corner.

are larger $O_3$ decreases in summer in the North China Plain and the Yangtze River Delta (Fig. 7c) than in other less-polluted regions. However, the overall magnitudes of surface $O_3$ decreases are not as large as simulated with UKESM1. There are noticeable differences in the latitudinal mean surface $O_3$ decreases, with the maximum changes estimated as 10 ppb in the bias

corrected simulation, smaller than 20 ppb predicted with UKESM1 (Fig. 7d). This indicates that the underlying impacts of emission controls on $O_3$ may not be as large as the model suggests, and that the $O_3$ responses to changing emissions may be overestimated. This is also reflected in the overestimation of $O_3$ changes in southern China in the SSP3-7.0 scenario (Fig. S4a, b, c).

In wintertime, while surface $O_3$ mixing ratios increase in high-emission regions under Net Zero, as seen in both UKESM1

and the bias-corrected results, areas with $O_3$ increases are smaller than those predicted by UKESM1 (Fig. 7). This again suggests that the magnitude and spatial extent of $O_3$ titration by NO may be overestimated in UKESM1. The same effect is seen in the bias-corrected wintertime $O_3$ under SSP3-7.0 (Fig. S4). In general, biases in $O_3$ simulations from UKESM1 are smaller in the Net Zero scenario but still remain large in the SSP3-7.0 scenario (Fig. S3b-d). These two scenarios correspond to low and high emission pathways, respectively, which indicates that the accuracy of $O_3$ simulations in UKESM1 may decrease

when emission changes become larger. The bias-corrected results show that only industrial regions with high $NO_x$ emissions in China show substantial $O_3$ increases under Net Zero, while surface $O_3$ mixing ratios decrease in less polluted regions in winter. This leads to a general decrease in latitudinal surface $O_3$ mixing ratios in wintertime (Fig. 7h).

With bias correction, the average surface $O_3$ mixing ratios are estimated to decrease in both seasons in the eastern part of China in the future under the Net Zero pathway. $O_3$ decreases of 5 ppb are predicted to occur in summer, which are slightly



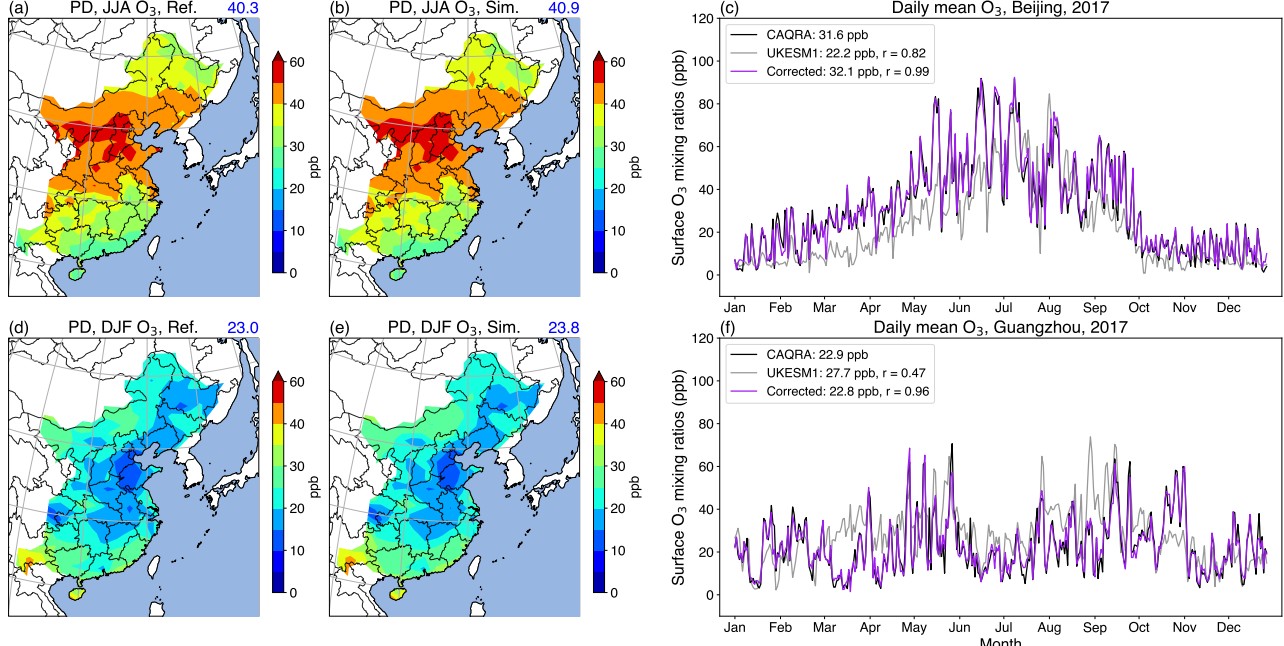

**Figure 6.** Surface mean $O_3$ mixing ratios derived from CAQRA (Ref.) in December-January-February (DJF) and June-July-August (JJA) over 2013-2017 **(a, d)**, compared with bias-corrected $O_3$ using deep learning **(b, e)**. Mean surface $O_3$ mixing ratios (ppb) are shown in the top right corner. Timeseries of daily mean $O_3$ mixing ratios in Beijing and Guangzhou in 2017 are shown in **(c, f)**, with mean $O_3$ values and correlation coefficients between CAQRA and the UKESM1 simulations and deep learning results shown in the legend.

larger than the 4 ppb decreases predicted in winter. This demonstrates the overall advantages of net zero policies in achieving a surface ozone air quality co-benefit. Furthermore, in high-emission regions, the directions of surface $O_3$ changes are different in summer and winter, as shown in both UKESM1 and the corrected UKESM1, indicating that VOC-limited $O_3$ formation still dominates there in winter.

We also calculate the annual average number of days with daily mean $O_3$ over 50 ppb as a measure to quantify high

$O_3$ pollution episodes, see Fig. 8. The number of days per year with high $O_3$ episodes under present-day conditions can be reproduced well following bias correction (Fig. 8a, b, Table 3), with intensive areas of high $O_3$ pollution in the North China Plain (60 days) particularly in summertime, and relatively low occurrence in the Pearl River Delta (31 days). There is an average of 33 days per year with high $O_3$ pollution over China. We find that the Net Zero policies will succeed in reducing the number of high $O_3$ pollution days markedly by 65 % in future. In contrast, following higher emission control policies will

increase high $O_3$ episodes by almost a factor of four (Table 3).

Following Net Zero emission controls, the Yangtze River Delta and the Pearl River Delta only have high $O_3$ episodes for a few days each year. However, high $O_3$ episodes still occur for almost one month (30 days) on the North China Plain and parts of central China in the future, demonstrating that $O_3$ pollution cannot be fully eliminated in this region. The Sichuan basin is




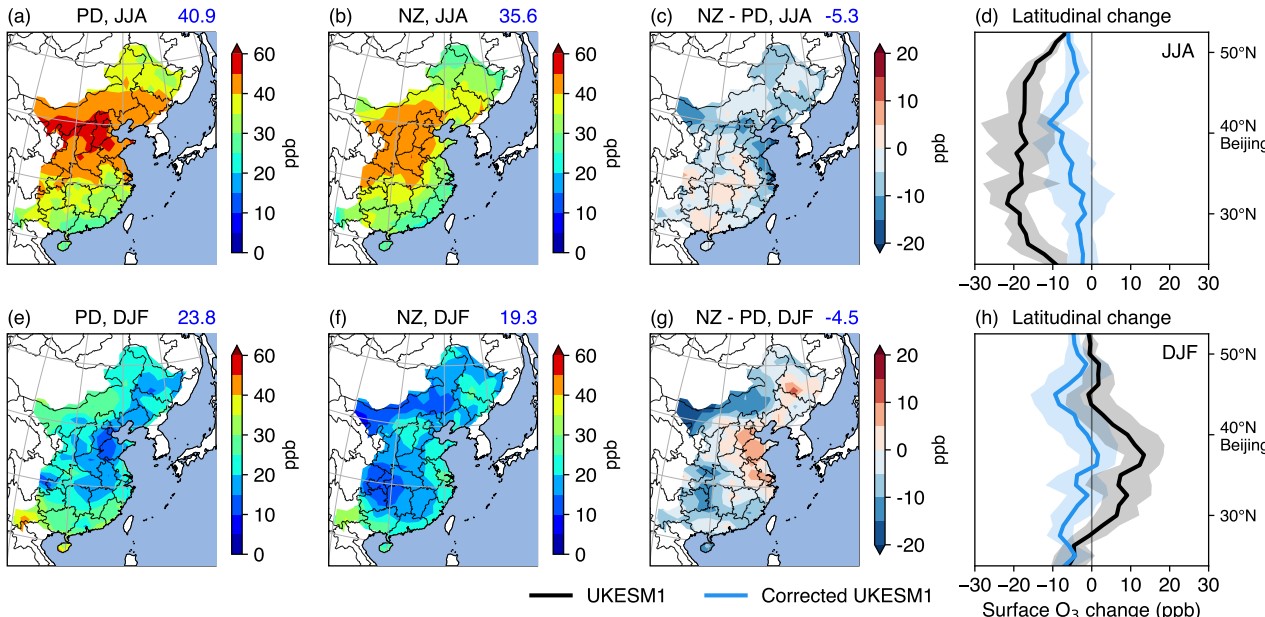

**Figure 7.** Seasonal mean surface $O_3$ mixing ratios corrected with the deep learning model in the present day **(a, b)** and the Net Zero scenario **(e, f)** in China, and the expected $O_3$ changes in summertime and wintertime **(c, g)**. Latitudinal mean $O_3$ changes in UKESM1 and bias-corrected UKESM1 are shown in **(d, h)**, where shading indicates one standard deviation of the changes in latitudinal $O_3$ mixing ratios.

also a region where high $O_3$ pollution cannot be fully addressed, likely due to the favorable meteorological conditions leading

to $O_3$ formation associated with the complex topography. Nevertheless, Net Zero policies are expected to deliver great benefits in mitigating $O_3$ pollution in China. Indeed, $O_3$ pollution is likely to become much worse if emissions continue to rise (Fig. 8d; Table 3). Even stricter controls on anthropogenic emissions than proposed to meet Net Zero may be required to avoid high $O_3$ pollution in the North China Plain.

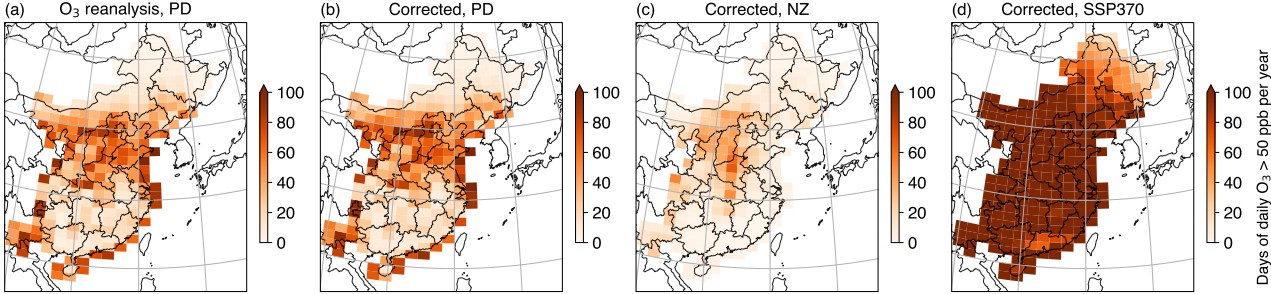

**Figure 8.** Annual average number of days with daily mean surface $O_3$ mixing ratios higher than 50 ppb in the present day calculated from **(a)** the surface $O_3$ reanalysis and **(b)** bias corrected UKESM1. Future high $O_3$ episodes under Net Zero **(c)** and SSP3-7.0 **(d)** pathways are shown from bias corrected UKESM1.





**Table 3.** Annual average number of days with daily mean surface $O_3$ mixing ratios higher than 50 ppb in China, the North China Plain (NCP), the Yangtze River Delta (YRD), the Pearl River Delta (PRD) and the Sichuan Basin (SCB). Conditions in the present day and under the Net Zero and SSP3-7.0 pathways are presented, calculated from the bias corrected UKESM1 simulations. The percentage change in the number of days in the future relative to the present day are shown.

| Number of days with daily mean $O_3 > 50$ ppb/Regions | Present day (Reanalysis) | Present day (Corrected UKESM1) | Net Zero (Corrected UKESM1) | SSP3-7.0 (Corrected UKESM1) |
| --- | --- | --- | --- | --- |
| China | 32.1 | 33.9 | 11.9 (-65 %) | 115.8 (242 %) |
| NCP | 56.9 | 60.5 | 30.6 (-49 %) | 123.7 (104 %) |
| YRD | 45.0 | 45.3 | 4.8 (-89 %) | 140.4 (210 %) |
| PRD | 31.2 | 31.4 | 1.6 (-95 %) | 117.0 (273 %) |
| SCB | 34.4 | 34.1 | 16.5 (-52%) | 139.3 (309 %) |

## 6 Conclusions

Net Zero emission polices are important for reducing regional surface $O_3$ pollution as well as for mitigating climate change. We use a chemistry-climate model to quantify the $O_3$ changes in China under a Net Zero pathway, and investigate the relative importance of different drivers of these changes. We also place our results in context by comparing to a scenario, SSP3-7.0 in which weak climate mitigation leads to continued increases in precursor emissions. Surface $O_3$ responses to Net Zero emission control policies in China are distinctly different in different seasons, with substantial $O_3$ decreases in summer and $O_3$ increases

in winter in high-emission regions due to decreased $O_3$ titration by NO. This demonstrates the large benefits of emission controls in reducing summertime average $O_3$ pollution in China by as much as 16 ppb.

Local emission changes are shown to be the most important driver influencing regional $O_3$ changes, which generally outweighs other drivers such as transport of $O_3$ from other countries, increased background $O_3$ formation through rising $CH_4$ abundance and a warmer climate. We do not find substantial changes in surface $O_3$ in China between Net Zero and SSP3-7.0

scenarios due to a warmer climate, but there are surface $O_3$ increases in southern China. Impacts of future local and external emissions on surface $O_3$ show strong seasonal variation, while increasing future $CH_4$ concentrations have a relatively uniform effect on $O_3$ throughout the year. In winter and spring, future external emissions outside China and higher $CH_4$ concentrations are more important than local emissions at a regional average scale.

We further demonstrate the capability of deep learning approaches to correct the biases in simulated daily mean $O_3$.

UKESM1 shows systematic biases in simulated $O_3$ like many other chemistry-climate models; these are expected to influence their projections of future $O_3$. Deep learning can provide improved assessment of the impacts of Net Zero policies on surface $O_3$. We find that surface $O_3$ changes are overestimated by UKESM1 in summertime, and therefore the benefits of emission controls may be overestimated by chemistry-climate models. UKESM1 estimates that the mean latitudinal surface



$O_3$ decreases due to emission controls could be up to 20 ppb in summer but bias correction shows that these may only be up to 10 ppb.

However, Net Zero emission policies succeed in reducing the number of days of high $O_3$ pollution by 65 % in China per year, with the number dropping from 33 days under present-day conditions to 11 days each year under Net Zero. The North China Plain will still be affected by high $O_3$ pollution in the future, meaning that stricter emission policies are needed in this region. In the Yangtze River Delta and the Pearl River Delta, $O_3$ pollution is likely to be less of a concern in the future as there are only a few days with high $O_3$ pollution under Net Zero. It is also clear that if emissions continue to rise, air quality in China will be substantially worse than at present, and therefore emission controls are essential. However, it is clear from these studies that emission controls can be very effective in reducing surface $O_3$ pollution, and that Net Zero emission policies can substantially mitigate $O_3$ pollution in China.

*Data availability.* The data generated in this study are available upon request.

*Author contributions.* All authors participated in designing the study. ZL conducted UKESM1 simulations, built the deep learning model, and performed the analysis with input and discussions from OW, RD, FO'C and ST. ZL, OW and RD prepared the paper, with contributions from all co-authors.

*Competing interests.* The contact author has declared that none of the authors has any competing interests.

*Acknowledgements.* Zhenze Liu, Oliver Wild, Ruth M. Doherty thank the project of the UK-China collaboration to optimise net-zero policy options for air quality and health (COP-AQ) under grants 2021GRIP02COP-AQ. Oliver Wild and Ruth M. Doherty thank the Natural Environment Research Council (NERC) for funding under grants NE/N006925/1, NE/N006976/1 and NE/N006941/1. Fiona M. O'Connor was supported by the Met Office Hadley Centre Climate Programme funded by BEIS and also acknowledges support from the EU Horizon 2020 Research Programme CRESCENDO (grant agreement number 641816). Steven Turnock would like to acknowledge support from the UK–China Research and Innovation Partnership Fund through the Met Office Climate Science for Service Partnership (CSSP) China as part of the Newton Fund.



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
