# Peer review of "Benefits of Net Zero policies for future ozone pollution in China"

_EGUsphere, 2023_

## Author Comment (AC1)

Dear editor and reviewers:

We thank the reviewers for their valuable comments which have been useful for the improvement of our manuscript. Specific concerns have been addressed below, with reviewers' comments shown in black bold, our responses in black and the added context in blue italic.

Responses to Reviewer 1:

1. **This manuscript simulated surface ozone changes in China under a Net Zero pathway using a chemistry-climate model which have corrected biases through deep learning model. The results indicated a substantial decrease in the monthly average surface ozone concentrations in the summer, with the greatest contribution from local emission reductions. The entire study appears technically sound, and the results are well interpreted. Thus, I recommend the publication by addressing the comments below.**

   We thank the reviewer for their positive comments here.

2. **For line 125, In addition to ozone, are there any other variables included as input in the deep learning model?**

   We use 20 independent meteorological and chemical variables as features, and in fact $O_3$ is not considered as an input variable in this study as its high correlation with $O_3$ biases would mask the contribution of other variables. We have now modified the text to explicitly list the variables considered to clarify this point.

   *Line 116:*

   *"We adopt the approach applied by Liu et al. (2022a) to use 20 physical, meteorological and chemical variables as features, and these include variables associated with location, season, temperature, humidity, wind speed, photolysis and deposition rates and concentrations of key precursors, see Liu et al (2022a). We do not use $O_3$ concentration as a variable, as this is highly correlated with $O_3$ biases and thus masks the contribution of other factors."*

3. **Why does methane contribute more significantly to ozone increase in western China compared to eastern China?**

   The magnitudes of surface $O_3$ changes largely depend on $O_3$ net photochemical production, and lower $O_3$ destruction in unpolluted regions may lead to a higher contribution of $CH_4$ to $O_3$ increases. We note that in clean high altitude regions e.g. the Tibetan Plateau, surface $O_3$ mixing ratios are naturally higher as they reflect mid-tropospheric ozone levels. We have now modified the text in section 4 "Drivers of future surface $O_3$ changes in China".

   *Line 167:*

*"Changes in atmospheric CH₄ abundance have a relatively uniform influence on surface O₃ in eastern China, with slightly greater effects in western China where altitudes are higher. A 4 ppb O₃ increase due to higher CH₄ is seen for both seasons (Fig. 2c, g)."*

4. **For figure 7d, the authors suggest that the significant smaller decreases of latitudinal mean surface O3 implies underlying impacts of emission controls on O3 may not be as large as the model suggests, and the overestimation of O3 responses to emission changes. However, by comparing figure 7 (a) and figure 1 (a), it can be observed that the current smaller decrease between the two scenarios after correcting with deep learning model is primarily due to a reduction in the corrected ozone concentrations. Does this imply that the decrease of ozone cannot fully demonstrate the reduction in pollution emissions?**

The reviewer is correct to note that the present-day surface O₃ concentrations in summertime simulated with UKESM1 are lower once the bias correction is applied. This is true as the O₃ simulated with UKESM1 is biased high in the present day. We have corrected for future conditions and find that the bias is smaller under these cleaner Net Zero conditions. The smaller difference between present day and future surface ozone in the corrected simulations simply tells us that the uncorrected UKESM1 simulations exaggerate the contribution of local emissions to surface ozone, and this is a key point that we make in the conclusions. The bias-corrected results give us a more reliable estimate of the response of O₃ to changing emissions and climate.

Responses to Reviewer 2:

1. **The authors quantified the future ozone improvement from the Net Zero policies by using the chemistry-climate model and deep learning model, and examined the different drivers of such changes in designed scenarios. I think this paper is interesting, and this topic of great importance to policy makers. However, several major revisions should be addressed before the recommendation.**

We thank the reviewer for their positive comments on this paper.

2. **First, unlike the PM₂.₅ pollution, ozone pollution is heavily affected by the meteorological variations from historical experiences (e.g., the period of 2013-2020), especially in the context of future warming climate. However, I didn't see the clear descriptions of meteorological data used for this study in the Method part, and how it changes over times and its effect and the interaction with climate change on future ozone pollution.**

We agree that meteorology affects surface ozone concentrations, and have already explored the effects of changing meteorology caused by climate change, see Fig. 2d, h. The model is driven by meteorological reanalysis data in the historical simulations (also called nudged simulations),

and is free running in the future scenarios. To clarity this, we have now added these details in section 2.1 'Description and application of the chemistry-climate model' as below.

*Line 84:*

*"...future simulations conducted as part of the Coupled-Model Intercomparison Project 6 (Eyring et al., 2016). We nudge the model with ERA-Interim meteorological reanalysis data for the present-day simulations, and allow the model to run freely in the simulations of future scenarios."*

The key aim of this study is to investigate the overall benefits of Net Zero policies on ozone mitigation in China. Since the reductions in anthropogenic emissions under Net Zero policies are large, the effects on surface ozone (Fig. 2a, e) considerably outweigh those from climate change. The effects of climate change between present day and 2060 are smaller than those between present-day and the end of the century, and are smaller under Net Zero policies than under any of the other SSP pathways. We find that the near surface annual mean temperature in East Asia is 17.2 °C in the present day, 19.7 °C under the Net Zero scenario (SSP 1-1.9), and 21.1 °C under the SSP3-7.0 scenario for 2060, and these changes are similar to those from the fully coupled UEKSM1 (Mulcahy et al., 2013) and CMIP6 model simulations (Tebaldi et al., 2020). Under high temperatures, ozone chemical formation is faster, and increased biogenic VOC emissions promote more ozone formation (Fig. S1), but increased humidity enhances ozone destruction (specific humidity in East Asia rises from 10.5 g/kg in the present day to 11.4 g/kg under the Net Zero scenario, and to 12.7 g/kg under the SSP3-7.0 scenario). All of these processes are represented in the chemistry-climate model used here, but we find that the overall influence of climate change on surface ozone remains far less than the effect of strong emission regulations, as expected.

3. **Second, is there any limitation or uncertainty by further applying the deep learning model (i.e., the correction ratio of historical results) to correct the biases in surface $O_3$ simulated with UKESM1 in future analysis?**

The main assumption we make in applying the deep learning model to future conditions is that that the driving variable values do not lie substantially outside the ranges associated with the present-day conditions used to train the model. Given that the climate changes are relatively small under the 2060 net zero conditions considered here, and that anthropogenic emissions are reduced in most places, we are confident that our results here will be robust. However, the reviewer is right to point out that this may not be the case in all scenarios, for example under SSP3-7.0 in 2100, when significant changes in climate and emissions may take the conditions outside the variable ranges used to train the model. This will introduce greater uncertainty, although it is not clear that it would lead to a systematic bias in the corrections. However, to address this point and others that the reviewer makes, we have now included a section on uncertainties in the paper.

Specifically, we have added discussion text regarding the underlying uncertainties in adopting this approach in section 5 "Bias corrected surface O₃ under the Net Zero pathway", and also added a new paragraph in the Conclusions section.

*Section 5, Line 213:*

*"...The evaluation demonstrates the capability of the deep learning model in correcting the seasonal and daily UKESM1 simulation of surface O₃. This approach shows great promise in reducing current model errors, and hence has the potential to improve simulations of surface O₃ under future scenarios.*

*Conclusions: Line 276:*

*"We acknowledge that there are uncertainties associated with the choice of deep learning model used and with the variables and parameters it is trained on, but the biases are sufficiently well predicted here that we are confident in the robustness of our results. The prediction might be further improved by employing more advanced deep learning architectures and considering a wider range of variables. The prediction of future surface O₃ biases may be slightly different under these conditions, but we believe that our principal results are likely to remain robust. The driving variables under the Net Zero scenario typically lie in the ranges associated with the present-day conditions that were used to train the model, suggesting that the relationships between inputs and outputs derived from the deep learning model are suitable for predicting future situations."*

4.  **Third, the authors should reorganize the result part, I cannot understand that why the "Bias corrected surface O₃ under the Net Zero pathway" part is presented at last in the results part as it is the basement of ozone improvement and drivers analysis, I think.**

We intentionally organized the structure of this paper in this way to highlight the differences between UKESM1 and UKESM1 with bias correction, and to caution that the assessment of environmental impacts with chemistry-climate models may be not accurate. Therefore, the first half of this paper demonstrates the benefits of Net Zero policies assessed with UKESM1 and the second half then highlights that these are not as large once bias correction is applied. Both these results are important, as we focus not only on the benefits of net zero policies on O₃, but also on the value of bias correction for generating more reliable estimates. To make the structure clear to readers, we have now stated the purpose of organizing the paper in this way at the end of the Introduction section.

*Line 61:*

*"In order to highlight the value of bias correction, we show the results of UKESM1 before showing the corrected results. We first investigate surface O₃ changes in China from the present day to the future under a Net Zero emission pathway simulated with UKESM1 in Sect. 3…."*

**5. Fourth, several limitations and uncertainties indeed exist in this study, for example, the model itself and meteorological data used, which should be systematically presented at the end of the manuscript.**

We acknowledge that there are uncertainties in the model and meteorology used, and while we have done our best to minimize these through use of a well-established model (UKESM1) and high-quality meteorological data for present day (ECMWF ERA), uncertainties will remain. These uncertainties are inherent in global model studies and are described in detail in previous assessments (e.g., Young et al., 2018). There are also uncertainties in the deep learning model applied here, and it is clear that the choice of parameters used may influence the results to some degree. However, from exploratory studies using other deep learning models we find that these generate very similar outputs to the method that we have applied here, suggesting that the results are robust. We have now included a section in the conclusions (see point 3 above) that discusses these uncertainties, and highlights that while improved methods may be available in future, we do not feel that these will alter the conclusions presented here.

**Minor comments:**

**6. Add quantitative results in the Abstract part.**

We have already included some quantification in the abstract, highlighting that summertime ozone changes are smaller after bias correction. We have now included some further quantification by adding a sentence on the number of days of high ozone episodes.

*Line 11:*

"*...may lead to a surface ozone decrease of 5 ppb in summer. The number of days with high ozone episodes with daily mean ozone greater than 50 ppb will be reduced by 65 % on average.*"

**7. Add more introduction of current ozone pollution condition in the first paragraph.**

We have now added some additional text in the introduction section to introduce the overall trend of surface ozone and ozone sensitivity regimes in China as follows:

*Line 25:*

"*...are less well regulated (Li et al., 2019). Observed summertime surface maximum 8 h average (MDA8) $O_3$ concentrations in China showed a consistent annual increase of 1.9 ppb between 2013 and 2019 (Li et al., 2020), and this increase is greater in high-emission regions, reaching 3.3 ppb per year on the North China Plain. Given that $O_3$ production in these regions tends to be VOC-limited (Wang et al., 2022), reducing emissions of $NO_x$ and VOCs simultaneously has become crucial.*"

**8. Please check and revise all the maps with 9-dash-lines in China.**

We intend the maps shown in the paper to be geographical and not geopolitical, and have revised them accordingly. Figs 1 and 2 show East Asia and not China, so we have rephrased the captions. Figs 6, 7 and 8 show the Eastern part of mainland China, which forms the principal focus of our study, and we have therefore corrected the caption to reflect this.

**9. Please explain the effect of external emissions outside China (like a circle) in Figure 2b (probably wrong, please check).**

'External' emissions in this study arise from countries outside China. These lead to areas of high ozone in countries surrounding China, but their effects are smaller over China itself. The transport of ozone from outside China explains the summertime gradient around China's borders, but has limited effects on central China where ozone from external sources makes little contribution. In winter this effect is less evident, as surface ozone concentrations in regions north of China are low due to reduced photochemical production. We have now added a sentence to reflect this in section 4 "Drivers of future surface $O_3$ changes in China".

*Line 167:*

*"...estimated here as 4 ppb (Fig. 2f).* *The contribution of external emissions is much larger near the country's borders than on central China."*

**10. Part 3: more quantitative results (i.e., numbers) should be added.**

We have amended the manuscript and added numbers for better descriptions of UKESM1 results in the section 3 of "Future surface $O_3$ changes in China under Net Zero policies".

*Line 146-147:*

*"... and the transport sector is shown to have the largest impact* *with 10 ppb $O_3$ increases."*

*Line 148-149:*

*"In wintertime,* *surface $O_3$ mixing ratios generally decrease by 1 ppb in the mainland China, but increase in eastern China by up to 20 ppb in heavily populated industrial regions."*

*Line 150-151:*

*"...power plants, industry and transport under SSP3-7.0 cause notable decreases in $O_3$ mixing ratios* *of up to 3 ppb* *in winter (Fig. S2e-g) although the effect of the residential sector is relatively small."*

**11. Figure 3, again, the small effect of climate on ozone pollution seems weird for me. Please add more explanation.**

We show the differences in surface ozone between SSP3-7.0 and SSP1-1.9 (Net Zero) scenarios due to differences in climate between these scenarios, and not differences between the present

day and SSP1-1.9 climate. We have stated this in the Abstract and in line 160 in section 4. The changes in temperature between these two scenarios are smaller than those between the present day and SSP1-1.9. We also note that the year of reference here is 2060 and not 2100, when changes would of course be larger. We have now modified the caption of Fig. 3 to make it clear.

*Caption Fig. 3.*

*"Seasonal surface O$_3$ changes relative to Net Zero due to changes in emissions in and outside China, global CH$_4$ concentrations and differences in 2060 climate under SSP3-7.0 in (a) China and (b) the globe."*

**12. As I know, generally model simulation has relatively bad performances on high ozone concentration (i.e., summer), while from Figures 6c and 6f, the model seems be not good in March, April (not the period of high values), please explain the reason.**

The reasons for ozone biases are different in different models, and the biases differ by region and by season. We note the presence of these systematic errors in section 2.3, "Development of the deep learning model". Surface ozone in UKESM1 is biased low in springtime over parts of northern China, and this is probably associated with excessive NO titration at this time of year, as emissions in the region are intense. This effect is stronger in coarse resolution global models than in finer resolution regional models which may be expected to capture spatial heterogeneity better. In the coastal region around Guangzhou, it is likely that the model biases are driven by meteorological influences associated with onshore and offshore flow, which may not be adequately resolved in the model. However, it is important to note that the cause of these biases is not important for our study, as the deep learning model captures their influence accurately during the bias correction process.

**13. There are many recent studies analyzing future ozone pollution in the context of carbon neutrality, please add the comparison.**

We have referred to other studies for comparison in the relevant sections as follows:

*Line 140:*

*"...demonstrating the great benefits of emission controls in mitigating summertime O$_3$ pollution. Other studies show similar results, with 18 ppb decreases in MDA8 O$_3$ mixing ratios achieved from Net Zero policies (Shi et al., 2021; Xu et al., 2022)."*

*Line 175:*

*"...with O$_3$ increases in the south and O$_3$ decreases in the north under a warmer climate. The regional differences are consistent with those found under the effects of changing BVOC emissions in future (Liu et al., 2022)."*

**Reference:**

Li, Ke, et al. "Increases in surface ozone pollution in China from 2013 to 2019: anthropogenic and meteorological influences." *Atmospheric Chemistry and Physics* 20.19 (2020): 11423-11433.

Liu, Song, et al. "Impact of Climate-Driven Land-Use Change on $O_3$ and PM Pollution by Driving BVOC Emissions in China in 2050." *Atmosphere* 13.7 (2022): 1086.

Mulcahy, Jane P., et al. "UKESM1. 1: development and evaluation of an updated configuration of the UK Earth System Model." *Geoscientific Model Development* 16.6 (2023): 1569-1600.

Shi, Xurong, et al. "Air quality benefits of achieving carbon neutrality in China." *Science of the Total Environment* 795 (2021): 148784.

Tebaldi, Claudia, et al. "Climate model projections from the scenario model intercomparison project (ScenarioMIP) of CMIP6." *Earth System Dynamics Discussions* 2020 (2020): 1-50.

Wang, Wenjie, et al. "Long-term trend of ozone pollution in China during 2014–2020: Distinct seasonal and spatial characteristics and ozone sensitivity." *Atmospheric Chemistry and Physics* 22.13 (2022): 8935-8949.

Xu, Beiyao, et al. "Impacts of regional emission reduction and global climate change on air quality and temperature to attain carbon neutrality in China." *Atmospheric Research* 279 (2022): 106384.

Young, Paul John, et al. "Tropospheric Ozone Assessment Report: Assessment of global-scale model performance for global and regional ozone distributions, variability, and trends." *Elem Sci Anth* 6 (2018): 10.

---

## Author Response (AR2)

Dear editor and reviewers:

We thank the reviewer for the comments which have been valuable for the further improvement of our manuscript. The comments are provided below in black bold, followed by our responses in black and the added context in blue italic.

**Responses to Reviewer 2:**

**The authors basically addressed the main concerns raised by the referees, and the manuscript has been significantly improved. While I still have one more concern according to the response (2) of the impact from meteorological variations in future.**

**Since the reductions in anthropogenic emissions under Net Zero policies are large, the contribution of future anthropogenic emissions will be small, and the effects of climate change on meteorological variations might be large. The authors should address the issue clearer.**

We thank the reviewer for their positive comments on our revisions to the manuscript. We note that the impact of large reductions in anthropogenic emissions from current levels to Net Zero masks most of the effects of climate change on surface ozone. The impacts of climate change are relatively small over the short time period considered in this study. Nevertheless, the reviewer is correct to point out that the relative contribution of climate change will become larger as emissions decrease to lower levels on the way toward Net Zero. Despite this, reducing anthropogenic emissions, relative to climate mitigation, is still the key to alleviating ozone pollution at the current time. Our conclusions therefore stand as they are currently presented. However, we have now included a sentence that reflects the reviewer's suggestion in our discussion in section 4.

Line 190:

*"… These $O_3$ increases occur in both seasons, but although they are more pronounced in summer, they remain much smaller than the changes due to anthropogenic emissions. The relative impacts of climate change on $O_3$ may become larger in future as anthropogenic emissions reduce towards Net Zero targets. Overall, we show that while local emissions are critical to $O_3$ pollution, …"*